# How Many Hungarian Consumers Choose Lactose- and Gluten-Free Food Products Even When They Do Not Necessarily Need to?

**DOI:** 10.3390/foods12213984

**Published:** 2023-10-31

**Authors:** Gyula Kasza, Erika Szabó, Tekla Izsó, László Ózsvári

**Affiliations:** 1Department of Applied Food Science, University of Veterinary Medicine Budapest, H-1078 Budapest, Hungary; kasza.gyula@univet.hu; 2Department of Veterinary Forensics and Economics, University of Veterinary Medicine Budapest, H-1078 Budapest, Hungary; szaboericska@gmail.com (E.S.); ozsvari.laszlo@univet.hu (L.Ó.)

**Keywords:** consumer behavior, consumer perception, lactose intolerance, celiac disease, gluten sensitivity, lactose-free food, gluten-free food

## Abstract

The popularity of “free-from” food products (FFFPs), which exclude several ingredients such as lactose, gluten, or sugar, is increasing globally. However, experts agree that avoiding these ingredients without medical reasons can lead to nutritional deficiencies. A representative consumer survey was conducted in Hungary (*n* = 1002); it focused on behaviors related to FFFPs, particularly lactose- and gluten-free products. This study revealed that consumers often consider “free-from” claims during shopping. Lactose- and gluten-free foods were popular, even among those without specific dietary needs. A distinct “free-from consumer group” (7.8% of the sample, predominantly women) was identified, who consume both lactose- and gluten-free foods frequently. However, only 15.4% of the group had medical reasons for their preference, such as lactose intolerance or gluten sensitivity. The majority (75.6%) chose these products without medical justification, relying on self-diagnosis, through the involvement of family members, or the belief that they were healthier. This consumer group accounts for nearly 6% of Hungary’s adult population, exceeding 470,000 individuals. Extrapolating these figures to other European countries suggests that 25–30 million EU citizens might be in a similar situation, highlighting the need for improved health education and awareness-raising campaigns to prevent imbalanced nutrition and foster the recognition and treatment of real health problems.

## 1. Introduction

A mindful selection of foods, following a particular diet, and healthy and sustainable eating habits have become and are still becoming more and more important for consumers nowadays. For example, nearly two-thirds of Hungarian consumers opt for foods without specific components or choose some kind of diet [1], which was found to be especially important for women, elderly consumers, and those who suffer from health problems, according to international and Hungarian research evidence [2,3,4,5]. As a result of global trends, the popularity of free-from and clean-label food products is constantly increasing worldwide [6,7,8]. These products are typically free of one or several ingredients, such as lactose or dairy, gluten, and palm oil, as well as GMO-free or free of additives (e.g., preservatives, food colors, etc.). The motivation behind these choices is usually supported by several factors: it may originate from health issues, food allergies or intolerances, risk-avoiding behavior, or even due to sustainability aspects for which the consumer may choose to avoid certain ingredients [6,7]. At the same time, subconscious heuristics and situational cues also influence consumers’ decision-making process about food purchases, which makes the reveal of motivations challenging [9,10].

Due to the growth of the demand, free-from foods are mainstream and easily available for all consumers [8,11]. Based on the trends of scarcely available data, which are already reflected in market revenues: sales of these products were estimated at over 2 billion USD in 2013 [8]. The market of free-from foods is mainly dominated by gluten-free and dairy- or lactose-free goods [11,12], providing the utmost sold products made for people with food intolerances and allergies [13]. In terms of gluten-free foods and beverages, North America had the largest market share, with 59% in 2012 [12], while in Europe, the UK and the northern countries have the most significant uptake per capita [11]. Although a gluten-free diet is essential for treating celiac disease, non-celiac gluten or wheat sensitivity, and wheat allergy, the main contributor to the expanding market of gluten-free products was the growing consumer group, who voluntarily adopted this diet without experiencing any adverse health effects from gluten [14]. Regarding lactose-free products, in Europe, Italy (EUR 772.9 million), Spain (EUR 440.9 million), and Finland (EUR 415 million) handled most of the market in 2017 [15]. This product category is now the fastest-growing market in the dairy industry, offering more and more products to consumers [16], and it is subjected to continuous innovation [17].

Looking at the rapidly evolving market trends, the following question may arise: what is the proportion of consumers who justifiably need gluten- and lactose-free products due to health issues, e.g., struggling with allergies or intolerances? Both conditions have a high incidence worldwide, with the inability or decreased ability to break down lactose being the most common malabsorption disorder [18], affecting about 400 million people [19], while celiac disease is one of the most common autoimmune diseases, of which prevalence is estimated to be around 1%, yet there are also research findings that assume 5–10% [20,21]. An inconsistency exists between the perceived and actual prevalence of food allergies and intolerances to common food items, and a significant number of consumers choose these free-from foods without medical evidence. In addition, family members of consumers suffering from food allergies and/or intolerances commonly follow the same diet against their own intentions if they live in the same household [22,23].

Avoiding the consumption of specific ingredients often raises long-term health risks, for example, due to nutritional issues such as vitamin and mineral deficiencies [24,25,26]. Moreover, consuming free-from products without justified necessity (medical diagnosis) also might be problematic, since several malabsorption and other gastrointestinal disorders have similar symptoms as a reaction to similar food categories, which can hinder the identification of the actual disease [27]. 

The various consumer preferences towards free-from food products raise some important research questions about the extent of the problem and the identification of consumer groups that are over-represented among unjustified “free-from eaters”. The present study aims to explore the situation in Hungary, focusing on the characterization of the concerned consumers and comparing them to the non-affected population. Four hypotheses were formed, reflecting the findings of previous research.

**H1.** 
*There are more women among the “free-from consumers”.*


**H2.** 
*The majority consume free-from foods as part of a healthy diet/lifestyle based on self-report.*


**H3.** 
*A significant proportion of Hungarian “free-from consumers” eat such products without medical examination, i.e., based on self-diagnosis.*


**H4.** 
*A significant proportion of Hungarian consumers consume such products due to health concerns of family members (e.g., the participant lives in a household with a person who has lactose intolerance or gluten sensitivity).*


The purpose of this study was to evaluate the extent of unnecessary “free-from” consumption (regular consumption without medical justification), with a focus on gluten- and lactose-free food products, and call attention to the problem of misperception of this product category.

## 2. Materials and Methods

The research was based on a quantitative consumer survey, for which personal interviews were conducted between July and August 2018 after a pilot testing period. In the pilot testing, a sample of 20 respondents was collected with a wide diversity of demographic variables. The analysis of the pilot testing suggested a few corrections to be made in the questionnaire before reaching the final version. The wording of the statements and questions was modified according to the feedback to make them easier to understand, especially for consumers who are less familiar with the subject. In addition, the length of the questionnaire was shortened in order to optimize the interview time. 

Answers from 1002 participants were collected according to a quota system ensuring proper representation of the Hungarian population in terms of sex, age, and NUTS-2 regions (NUTS-2: Nomenclature of Territorial Units for Statistics) (Table 1). The quota system was defined based on the actual micro-census conducted in 2016 by the Hungarian Central Statistical Office. The 15–20 min long interviews were performed in public locations in twelve cities of different sizes (Budapest, Dombóvár, Eger, Füzesabony, Győr, Kiskunfélegyháza, Miskolc, Siófok, Szeged, Székesfehérvár, Szolnok, and Veszprém) covering the whole country. 

The topics of the questionnaire focused on the aspects of food product choices (e.g., considered product attributes, labels), knowledge about products, health issues of the participants (diseases linked to nutritional and dietary aspects), shopping habits (e.g., location of purchase), and socio-demographic subjects such as level of income and education. Information related to the health status of the participants was recorded based on self-declaration, the survey did not include clinical tests or require the presentation of a medical verification. The questionnaire principally consisted of close-ended questions using the 5-point Likert scale system (1: strongly disagree; 5: strongly agree). All the included statements and questions in the mentioned topics were formed based on previous research [28,29,30]. The questionnaire is enclosed as Appendix A. The frequencies of general socio-demographic characteristics of the sample are presented in Table 2.

As for the characterization of the results, descriptive analysis was conducted by the IBM SPSS 25.0 software [31] from the dataset filtered for errors. Cronbach’s alpha coefficient was used to check the reliability of the variables in the questionnaire, which showed acceptable results (overall, Cronbach’s α = 0.949 for the 196 items). Crosstab statistics with Pearson’s Chi-square test supported by the z-test and the Mann–Whitney U test were also applied. The confidence level was set to 95% for each test.

In order to receive a clear distinction between segments of consumers, a robust sample was collected, but only a relatively low number of variables were used in this study. Answers were collected from the respondents in regard to 8 “free-from” categories; however, this study only focuses on gluten- and lactose-free food products. The reason for this decision is that these two traits are completely unnecessary to those consumers who do not suffer from any kind of gluten sensitivity or lactose intolerance, while other “free-from” claims might deliver benefits to a more general group of consumers. More complex statistical analytical methods, such as factor analysis, cluster analysis, and structural equation modelling would allow the exploration of a more detailed structure behind consumer decisions, but that is out of the scope of the present study.

## 3. Results and Discussion

The results confirmed that free-from claims on labels are usually considered by consumers during shopping as an influencing factor (Figure 1). The highest Likert scores were obtained by the “sugar-free” label (3.19 ± 1.44 out of 5), followed by “free from saturated fats” (2.76 ± 1.27 out of 5) and “fat-free” (2.54 ± 1.27 out of 5). Lactose- and gluten-free claims reached values below 2.5 (2.41 ± 1.38 and 2.30 ± 1.32, respectively), suggesting that these product characteristics are of interest only to a smaller group of consumers. However, generally low scores may also indicate that respondents pay little attention to the labels when shopping, routinely choosing products and buying the same products repeatedly [32].

The questions on the frequency of consumption of such products were in line with the previous topic on free-from labelling: the most frequently consumed products from the list were carbohydrate-free or low-carbohydrate foods (2.76 ± 1.31 out of 5) and sugar-free foods (2.74 ± 1.41 out of 5). The frequency of consumption of lactose- and gluten-free foods for the whole sample (*N* = 1002) was found to be lower but still prevalent in the sample (2.05 ± 1.34 and 1.90 ± 1.20 out of 5, respectively). Based on statistics of lactose-free food consumption frequency in Germany, the proportion of people consuming such products on a daily basis is also the lowest in the population, but it doubled between 2018 and 2021 [11]. In the case of gluten-free foods, the amount consumed also shows a continuous increase; for example, in Italy, the amount of gluten-free food produced increased by almost 2.5 times in the interval 2010–2015 [33].

Compared to other “free-from” categories, lactose-free and gluten-free foods do not offer any health benefits to consumers without specific dietary needs. For them, these products are just a more expensive alternative compared to traditional food-stuffs and can even deliver nutritional deficiencies in longer periods without specific supervision. However, there is a remarkably great sized group (n = 78)—hereafter named the “free-from group” or “free-from consumers”—whose members consume both lactose- and gluten-free foods particularly frequently (4.51 ± 1.36 and 4.44 ± 1.20 out of 5).

For further analysis, this so-called “free-from consumers” group (n = 78) was compared to the “ordinary consumers” group (n = 924) in terms of demographic indicators, as well as lifestyle and nutrition-related attitudes. Age, income, and educational level were not significantly different between or within the “free-from” and “ordinary consumers” groups according to the χ^2^- and z-tests. However, as hypothesis H1 assumed, there is a difference regarding sex: there are significantly more women in the “free-from consumers” group (χ^2^ = 6.17; df = 1; *p* = 0.013). The proportion of women in the “free-from consumers” group is 66.7%, compared to 52.1% in the “ordinary consumers” group. The difference can also be considered significant within the “free-from group” (Table 3).

This result, therefore, not only confirms hypothesis H1 but is also consistent with previous research in the field, which found that women tend to purchase a higher proportion of free-from products [34].

The same correlation was observed concerning healthy eating and living a healthy lifestyle. In the “free-from consumer group”, more respondents paid attention to these aspects, and their attitudes towards them were also confirmed to be significantly higher (Table 3). The “free-from consumer group” is not only more interested in a healthy diet and lifestyle but also makes more conscious choices when buying food, and the quality of the food is also essential (4.48 on the 5-point scale) for them. Being conscious during shopping and reading more carefully about the products’ labelling can be linked either to the avoidance of specific ingredients or to a healthy lifestyle in general [35]. Consumers in the “free-from group” are willing to pay more for foods considered to be “healthy”, and this product attribute is significantly more vital for them than to the ordinary consumer group (Table 4). This suggests that hypothesis H2 has also been confirmed, as the “free-from consumer group” assumingly not only prefers lactose- and gluten-free foods but usually considers them as part of a healthy diet.

Besides consumption patterns, it is also interesting whether choosing lactose- and gluten-free products is reasonable in a medical aspect or not. Regarding lactose intolerance, 37.0% of the “free-from consumers” reported having enzyme deficiency, 15.1% only had a family member struggling with the problem but they were not affected directly, while 47.9% consumed such products without being affected at all. In the case of gluten-free consumers, only 25.4% of the “free-from group” responded positively about having a health issue, 19.7% of them had a family member being affected in the household, and 54.9% of the group did not have any reasons to follow a gluten-free diet that could be explained by direct or indirect effects. In a household survey conducted in the United Kingdom, 20% of the population claimed to have food intolerances, but based on double-blind placebo-controlled food challenges, less than 2% had real reactions to certain foods. Similarly, in a study conducted in Germany, a questionnaire revealed that one-third of respondents experienced food-related reactions, but subsequent double-blind placebo-controlled tests identified the actual prevalence of adverse food reactions at 3.6% [36,37]. In 2019, a survey conducted in Italy found that 6.9% of men and 10% of women who were lactose-intolerant consumed lactose-free products, while 15.6% of men and 19.4% of women were non-sensitive yet usually consumed such foods [38]. According to Axelsson et al. [39], 11% of American households purchased gluten-free foods, which is 10% more than the real ratio of the population that is sensitive to gluten. Still, as a seemingly healthy diet, many people adopt it without having symptoms [40]. Based on a study in 2016, 22% of Hungarian households bought gluten-free products that year, with only 2.5% mentioning the gluten sensitivity of a family member. Also, 86% of those who purchased lactose-free foods in Hungary made this choice without lactose digestion problems [41].

The question about the verification by medical examinations details the picture further. The results show that most of the “free-from group” did not consult a doctor about the need for lactose-free or gluten-free foods; 78.5% of the group had not been tested for lactose intolerance, while the ratio was 77.3% for sensitivity to gluten or coeliac disease. These ratios seem to be slightly better than the results of a Hungarian study conducted in 2016 about possible health risks of the unreasonable free-from diet [41]. Table 5 summarizes the justification for following a free-from (lactose- and/or gluten-free) dietary regime, indicating both personal- and family-level involvements combining lactose and gluten avoidance.

The results in Table 5 confirmed the hypothesis on the consumption of free-from foods without medical examination (H3) and about following a free-from diet related to a family member (H4); 35.9% of the “free-from group” consume lactose- and/or gluten-free foods without any symptoms or having a family member affected by any of the health issues. Although living in the same household as a family member who must follow a lactose- and gluten-free diet does not verify the consumption, which is nevertheless risky, it can still be considered an attenuating circumstance. Only 15.4% of the “free-from consumers” have to deal with both lactose and gluten intolerance or sensitivity that fully justifies their free-from dietary preference. It also highlights that at least 75.6% of the “free-from consumers” do not have any justification for this dietary pattern: from the 100.0% “free from consumers”, 15.4% had medical justification for both issues and 9.0% were missing answers. These results are in parallel with the findings of a previous Hungarian study showing that among the persons who declared themselves to be lactose sensitive, around half of the women and one-third of the men were medically proven lactose sensitive, so more than half of the respondents declared themselves affected on basis of self-diagnosis without any medical confirmation [27].

Consuming free-from products without medical confirmation and only relying on self-diagnosis can be problematic in several respects. The issue can originate from multifarious and very different conditions (e.g., gastrointestinal symptoms and abdominal pain present in lactose intolerance, histamine intolerance, irritable bowel syndrome, etc.), which may be triggered by the same product category such as dairy [36,42]. For example, although hypolactasia, or symptomatic lactose malabsorption, can be effectively detected, the identification of lactose intolerance rather often relies on self-diagnosed symptoms [27,43]. The problem with self-diagnoses and voguish dietary guides (e.g., promoting a free-from diet) is that they are administered without professional control or the necessary medical pre-examinations [44]. The occurrence of self-diagnosis is also confirmed by questionnaire-based data, which shows that the prevalence of food allergies has increased, whereas the number of cases with a medical diagnosis has not [45]. Nevertheless, the absence of reliable diagnostic biomarkers for various food intolerances also makes it difficult to identify specific problematic foods for individuals. The trial-and-error method can be useful, where suspected food components are temporarily reduced and then reintroduced to the diet while assessing the reaction. Hence, the lack of proper diagnostic tools contributes to the wrong estimation of the prevalence of food intolerances, but this can only explain a small portion of the disparity since only a small fraction of self-reported food sensitivities can be medically confirmed in terms of anatomical and pathological differences in the human body. Many people are, therefore, incorrectly attributing their symptoms to food sensitivities (both allergies and intolerances). Several factors contribute to this misdiagnosis, including psychological factors such as confusion regarding the diagnosis, coincidental associations between food and symptoms (e.g., observing physical signs when consuming food during emotional distress), psychosomatic responses, and taste aversions. Additionally, there are biological mechanisms not fully explored in this research field, such as the conditioning of the immune system or stress-related responses, which may be relevant [36,46,47]. Thus, whatever reasons are behind the roots (psychological or physical), accurate identification is necessary, as a diagnosis not confirmed by a physician is risky for two reasons. On one hand, an incorrect self-diagnosis may hinder the recognition and treatment of a real health problem. On the other hand, an unjustified free-from diet may lead to imbalanced nutrition (and unnecessarily high costs—specific products, such as free-from, functional, and vegan products, are usually more expensive than others). The omission of dairy products from the diet, for example, can lead to the development of severe calcium and vitamin D deficiency [48]. Although the results of the studies are contradictory [49,50], it is still widely believed that lactose intolerance is a risk factor for osteoporosis due to decreased calcium intake caused by the abandonment of dairy products [18,49,51]. In regard to a gluten-free diet, it is also linked with possible drawbacks such as a decreased intake of minerals (e.g., calcium, iron, magnesium, zinc), vitamins (vitamin B12, folates, vitamin D), and fibers, while the exposure of consumers to arsenic can be higher [39]. Additionally, a gluten-free diet often includes foods with increased levels of hydrogenated and saturated fatty acids and a higher glycemic index [39] and a tendency for consuming dishes rich in fat, sugar, and calories [12]. Without medical supervision, therefore, a gluten-free diet can lead to nutritional deficiencies and elevate the risk of constipation [39].

Free-from products usually stand out from other foods on the shelves because of their different labelling, as they are supposed to inform consumers with allergies or intolerances linked to certain ingredients (although, labelling related to all intolerances is not necessarily obvious due to the lack of EU regulation about the definition and usable claims of lactose-free products [52]). However, research confirmed that people tend to interpret food labelling in a particular way, often not related to its original purpose, and may feel encouraged to buy free-from goods, as they perceive a healthier product image [28,53,54]. Regarding nutritional labels, a study also pointed out that consumers might not engage in reasoned thinking when encountering ‘front of pack’ labels, but rather rely on decision heuristics [55]. It should also be noted that studying the motivations behind food choices and consumer behavior through self-reporting surveys can be challenging due to several reasons related to the sub-cognitive bias, heuristics, and situational cues in decision-making. For example, the rationalization of food choice by attributing them to conscious and socially acceptable factors, being unable to recall the specific influence, or giving responses aligning with cultural norms rather than admitting the irrational factors, impacts originating from the environment, etc., all can have significant effects on behavior [9,10,56].

In addition to the specific diet or lifestyle a consumer follows, the preference for clean-label or free-from products greatly depends on the consumer’s general knowledge of nutrition [28], which is often based on incomplete or incorrect information [51]. Moreover, a cognitive bias (in fact a cognitive shortcut) can be observed. The healthiness of food products is sometimes evaluated by consumers on the basis of their category membership, and category membership is based on the claims on the label. However, the consumers are often not able to interpret the meaning of the differentiating claims [28,55]. This is verified by the survey implemented by YouGov [57], pointing out that free-from products are considered to be less processed, have lower fat, sugar, and salt contents, contain more vitamins, and are more organic. Moreover, gluten and lactose were perceived as harmful ingredients. Otherwise, in order to prevent and avoid diseases, generally women and the elderly are more likely to choose free-from foods that are considered to be healthier [58,59,60].

There may be deeper-rooted psychological reasons for preferring free-from products in addition to those mentioned above. The consumption of food free of various substances and considered to be healthier may also be explained by an eating disorder identified a few decades ago. Because of “healthy food addiction”, called orthorexia nervosa, the consumer develops extreme dietary habits based on “excessively healthy eating”, which can be traced back to a previous illness, or to the dietary changes it made necessary in the past [61]. Recently, Savarese et al. [62] observed during the COVID-19 pandemic that for anxious people, consuming free-from foods fosters a feeling of regaining control over their lives.

## 4. Conclusions

This study’s results show that the “free-from customers”, who consume both lactose- and gluten-free foods particularly frequently, are mostly women, and they pay more attention to their own and family members’ health and are willing to pay more to stay healthy and pursue a healthy lifestyle. That is, these customers are more prone to buy free-from products, especially carbohydrate- and sugar-free food, than the “ordinary customers”. This is supposed to be because in Hungary, most women do the shopping themselves, while most men prefer shopping together with other family members; thus, women can be regarded as decision-makers in food procurement. Only a small minority, 15.4% of the “free-from costumers”, have both lactose and gluten intolerance or sensitivity, and more than three-quarters of this group were not tested medically for either disorder; thus, a significant proportion of this group consumes the free-from products based on self-diagnosis, through the involvement of their family members, or simply by following the belief that they are a more healthy alternative to traditional food products. Their number stretches to about 6% (7.8% of the population belongs to the “free-from consumer group” multiplied by 75.6%, which is the ratio of those who did not have justification for it) of the Hungarian population, which is more than 470.000 adult people. Assuming that, this ratio has relevance in other European countries, we might conclude that 25–30 million citizens might be in the same situation in the European Union only. These findings revealed information gaps that could be addressed by raising awareness of the importance of medical examination through the education of both children and adults in order to avoid long-term health risks derived from nutritional deficiencies.

## Figures and Tables

**Figure 1 foods-12-03984-f001:**
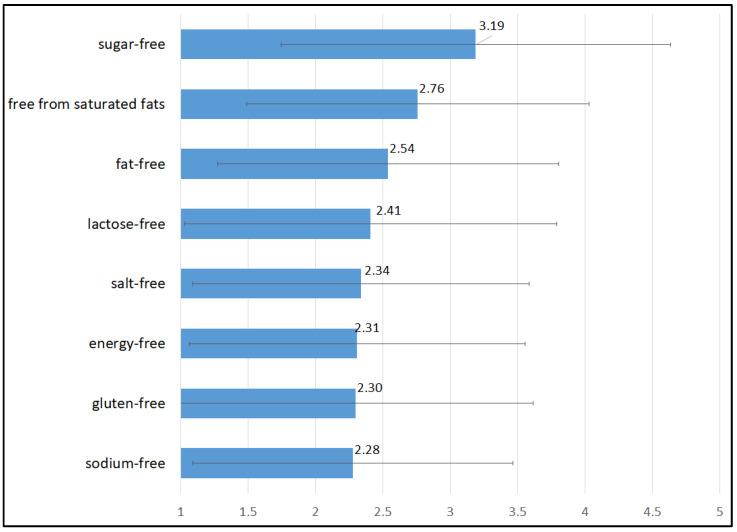
Importance of free-from claims on the label during food shopping (1: strongly disagree; 5: strongly agree).

**Table 1 foods-12-03984-t001:** Demographical characteristics of the sample compared to the Hungarian population (n = 1002).

Variable	Category	Ratio in the Sample (%)	Ratio Based on the Census of 2016 (%)
Sex	Female	53.20	53.07
	Male	46.80	46.93
Age	18–29 years	18.00	17.59
	30–39 years	17.00	17.04
	40–59 years	34.60	33.83
	60 and above	30.50	31.54
Region	Central Hungary incl. Budapest	31.04	30.75
	Central Transdanubia	10.68	10.80
	Western Transdanubia	10.18	10.03
	Southern Transdanubia	8.68	9.13
	Northern Hungary	11.68	12.62
	Northern Great Plain	15.17	14.90
	Southern Great Plain	12.57	12.78

**Table 2 foods-12-03984-t002:** Type of residence, education, and income levels of participants (n = 1002).

Variable	Category	Ratio in the Sample (%)
Type of residence	Village	15.27
	City	60.68
	Capital	22.65
Education	Primary education	2.00
	Secondary education	42.12
	Higher education	54.29
Level of income	Very low	0.70
	Low	11.88
	Average	65.37
	Above average	16.47
	Outstanding	1.50

**Table 3 foods-12-03984-t003:** Crosstab of the consumer groups in terms of sex (letters in superscripts indicate the result of the z-test).

Consumer Groups		Sex		Total
		Female	Male	
Ordinary consumers	N	481 ^a^	443 ^b^	924
	Ratio in the group	52.1%	47.9%	100.0%
Free-from consumers	N	52 ^a^	26 ^b^	78
	Ratio in the group	66.7%	33.3%	100.0%

**Table 4 foods-12-03984-t004:** Mean ranks and significance of the difference between the two consumer groups based on the Mann–Whitney U test.

Statements	Free-from Consumers (*n* = 78)		Ordinary Consumers (*n* = 924)		Mann-Whitney U	*p*
	Mean	Mean Rank	Mean	Mean Rank		
I am constantly learning about food and nutrition.	4.17	634.76	3.57	483.59	24,705.50	<0.01
I read the product labels carefully.	4.09	630.26	3.50	485.80	25,005.00	<0.01
It is important for me to eat healthy.	4.45	580.93	4.27	488.85	28,649.50	0.003
I am willing to pay more for a “healthy food”.	4.25	579.45	3.93	486.87	28,124.00	0.004
I am interested in healthy eating.	4.46	587.62	4.20	487.07	28,304.50	0.001
I am interested in scientific questions about lifestyle.	4.18	585.85	3.79	487.77	28,521.00	0.002
Food quality is the most important consideration when I am shopping for food.	4.47	558.35	4.34	491.92	30,183.50	0.031
The nutritional composition of foods is the most important consideration when I am shopping for food.	4.18	656.11	3.51	481.95	22,783.50	<0.01
Fitting into a healthy diet is my main consideration when I am shopping for food.	4.08	679.09	3.33	482.60	21,638.00	<0.01

**Table 5 foods-12-03984-t005:** Cases of the justification for the consumption of lactose- and/or gluten-free food products.

Justification	Number in the “Free-from Group”	Ratio (%)
No justification at all	28	35.9
Concerned in one health issue, no family member affected	10	12.8
Affected by both health issues personally	12	15.4
One family member is affected, but the respondent is not at all	8	10.3
Concerned in one issue and one family member is involved in another issue	9	11.5
Family members are affected by two health issues, but the respondent is not at all	4	5.1
Missing answers	7	9.0
Total	78	100.0

## Data Availability

The data presented in this study are available on request from the corresponding author.

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
