# Peer review of "How Many Hungarian Consumers Choose Lactose- and Gluten-Free Food Products Even When They Do Not Necessarily Need to?"

_foods, 2023, doi:10.3390/foods12213984_

Round 1
Reviewer 1 Report
Comments and Suggestions for Authors
A representative consumer survey was conducted in Hungary in the manuscript entitled "How many consumers prefer lactose- and gluten-free food products even when they don't necessarily need to?" (Manuscript ID: foods-2638786) submitted to Foods. It focused on behaviors related to "free-from" food products, particularly lactose- and gluten-free products. This study is interesting for governments in monitoring and regulating their residents' consumption behavior, especially in regions where "free-from" food products are prevalent in recent years. The authors also interpreted the possible disadvantages of unnecessary food consumption and feasible interventions. While the manuscript was well designed, a few issues should be addressed. The main concerns:
- Data was obtained only from a survey in Hungary. Thus, Hungary should be included in the title to help limit the region of this study and make it more accurate.
- There are some things that could be improved in the form of citations in the manuscript, and careful revision of them should be made.
- Long-term health risks caused by consuming "free-from" food products by people who do not suffer from health issues would be the main reason for this study. However, this information was not described sufficiently in the introduction.
Author Response
Thank you for reviewing our manuscript and your contribution to the improvement through your suggestions.
Our responses are provided by point-by-point:
Comment 1.1:
- Data was obtained only from a survey in Hungary. Thus, Hungary should be included in the title to help limit the region of this study and make it more accurate.
Answer 1:
Thank you for the remark, we agree with your suggestion. The new title now includes „Hungarian consumers”.
Comment 1.2:
- There are some things that could be improved in the form of citations in the manuscript, and careful revision of them should be made.
Answer 1.2:
We revised the references according to your recommendation, removed the ones that were not necessary and added new ones where it was needed. However, amendments made in the reference list cannot be seen with “track changes” due to some technical issues with the numbers of the cited materials, we apologize for the inconvenience.
Comment 1.3:
- Long-term health risks caused by consuming "free-from" food products by people who do not suffer from health issues would be the main reason for this study. However, this information was not described sufficiently in the introduction.
Answer 1.3:
Thank you for the remark, we dedicated a short paragraph to this topic in the Introduction per your suggestion. A more detailed explanation of the subject is in the Results and Discussion section since the Introduction would have been too long.
Reviewer 2 Report
Comments and Suggestions for Authors
The topic of the presented work is very current and very interestingly processed. Nevertheless, I have comments about the work:
Introduction:- it would be appropriate to shorten it. The paragraph from L65 to L91- should be moved to the discussion.
Discussion: To discuss the obtained results more with global findings, especially taking into account the EU. It would be appropriate to add data about how the worldwide consumption of lactose- and gluten-free foods has increased over the years.
Author Response
Thank you for the constructive review, we appreciate your comments.
Our responses are provided point-by-point:
Comment 2.1:
Introduction:- it would be appropriate to shorten it. The paragraph from L65 to L91- should be moved to the discussion.
Answer 2.1:
Thank you for the suggestion, we moved the mentioned paragraph to the Results and Discussion section.
Comment 2.2:
Discussion: To discuss the obtained results more with global findings, especially taking into account the EU. It would be appropriate to add data about how the worldwide consumption of lactose- and gluten-free foods has increased over the years.
Answer 2.2:
We completed the Results and Discussion section with some data on the growing lactose- and gluten-free food consumption in European countries based on your suggestion.
Reviewer 3 Report
Comments and Suggestions for Authors
Title, abstract, introduction
1. Title: change according to the aims of the study, here the topic is not preference
2. Abstract: describe what exactly is "free-from consumer group"
3. In the manuscript, provide evidence how the subjects proved their status regarding diagnosis of lactose intolerance and gluten sensitivity/celiac disease
4. H4: ‘’through the involvement of their family members’’: what does this mean? Please provide alternative wording
Materials and methods
5. How long the interview lasted?
6. What specific changes were made through pilot testing?
7. Provide the questionnaire as the supplementary material
8. Provide background for forming statements where Likert scale was applied
9. Level of income was self-evaluated? If so, this is a problem, more objective numerical categories should be used
10. Level of education: very simplified, what is the official categorization in Hungary?
11. Discussion instead of discussions; please check other typing errors throughout manuscript
12. Figure 1: energy-free is inappropriate term, please change
13. What do you mean exactly with ‘’Food quality’’? indicate parameters
14. Ref 46: check if it relevant to the statement in the text, if not necessary and representing national data, replace with article in English; check all other references
Author Response
Thank you for your thorough review and the constructive recommendations.
Our responses are provided point-by-point:
Comment 3.1:
Title, abstract, introduction
- Title: change according to the aims of the study, here the topic is not preference
Answer 3.1:
Thank you for the constructive remark, we have changed the word „prefer” to „choose”, which better reflects the topic.
Comment 3.2:
- Abstract: describe what exactly is "free-from consumer group"
Answer 3.2:
Thank you for noticing, we added a short explanation to the Abstract about the „free-from consumer group” as you suggested.
Comment 3.3:
- In the manuscript, provide evidence how the subjects proved their status regarding diagnosis of lactose intolerance and gluten sensitivity/celiac disease
Answer 3.3:
Both diagnoses were recorded based on self-reporting, the survey did not require any proof of medical examination. Based on your comment, we included this information in the Methodology section of the manuscript.
Comment 3.4:
- H4: ‘’through the involvement of their family members’’: what does this mean? Please provide alternative wording
Answer 3.4:
Thank you for your comment, we changed the wording of the hypothesis to a clearer form: „A significant proportion of Hungarian consumers consume such products due to health concerns of family members (e.g. the participant lives in a household with a person who has lactose intolerance or gluten sensitivity).”.
Comment 3.5:
Materials and methods
- How long the interview lasted?
Answer 3.5:
The interviews generally lasted for 15-20 minutes. We added this information based on your remark, thank you for noticing.
Comment 3.6:
- What specific changes were made through pilot testing?
Answer 3.6:
We have changed the wording of the statements and questions to make them easy to understand even for consumers who are less familiar with the subject, and in addition, we have shortened the length of the questionnaire in order to optimize the filling time. Thank you for noticing this shortcoming, we completed the methodology section according to your comment.
Comment 3.7:
- Provide the questionnaire as the supplementary material
Answer 3.7:
As per your comment, we enclosed the questionnaire as supplementary material.
Comment 3.8:
- Provide background for forming statements where Likert scale was applied
Answer 3.8:
Thank you for the meaningful observation on the missing references, we have cited all the research, which were studied during the preparation of the questionnaire.
Comment 3.9:
- Level of income was self-evaluated? If so, this is a problem, more objective numerical categories should be used
Answer 3.9:
The income level was based on the self-evaluation of the respondents about their household’s situation. In previous surveys, we tried several types of numerical categories and intervals, but respondents often skipped this question, because they considered it confidential information. Here we have to remark that Hungarian people are generally very shy about disclosing information on their earnings. The categorization used in this study is a well-established methodology for assessing the participants' financial possibilities in our experience, and able to gather far more answers compared to when we asked for an amount per person in the household or a numerical category to be chosen. We also learnt from the literature that even if a numerical categorization had been used, the estimation would not show the respondents' realistic income level because of the hedonic recall bias (Prati, A., 2017. Hedonic recall bias. Why you should not ask people how much they earn. Journal of Economic Behavior & Organization, 143, pp.78-97.). It should also be noted that the study did not aim to be representative of the Hungarian population in terms of this variable, only for age, sex and geographical distribution.
Comment 3.10:
- Level of education: very simplified, what is the official categorization in Hungary?
Answer 3.10:
The official categorization is:
|
Type of public education institution in Hungary |
ISCED level |
Category in the research |
|
Elementary school |
1-2 |
Primary education |
|
Vocational high school, Technical school |
3-5 |
Secondary education |
|
High school, Grammar school |
3-4 |
Secondary education |
|
College, bachelor's degree |
6 |
Higher education |
|
University, master's degree |
7 |
Higher education |
In the questionnaire, „secondary education” was indicated by two variables: „Vocational high school or technical school” and „Having a high school diploma” (and not as e.g. Grammar school) as the highest educational level. The Hungarian system makes it possible to finish vocational and technical education with or without taking the general final school-leaving examination (which exam is similar to the European Baccalaureate), which provides a „High school diploma”. However, in high schools and grammar schools, taking this final exam and getting the diploma is part of education in all cases. But both of these education types are at the secondary level, therefore these two variables were merged for the analysis. The college and university degrees were indicated as 1 variable in the questionnaire, since these are both above the „high school diploma” level and considered as „higher education” by the population.
Comment 3.11:
- Discussion instead of discussions; please check other typing errors throughout manuscript
Answer 3.11:
Thank you for your remark, we have checked the whole manuscript and corrected the typing errors.
Comment 3.12:
- Figure 1: energy-free is inappropriate term, please change
Answer 3.12:
You are completely right because completely “energy-free” foods are really scarce in the market. We have decided to use this term in the questionnaire because there are several food products labelled this way in Hungary and other European countries. This term has been also justified by Regulation 1924/2006 which sets the legal framework for nutrition and health claims made on foods in the European Union. It allows the term to be used the following way: “Energy-free: A claim that a food is energy-free, and any claim likely to have the same meaning for the consumer, may only be made where the product does not contain more than 4 kcal (17 kJ)/100 ml. For table-top sweeteners, the limit of 0,4 kcal (1,7 kJ)/portion, with equivalent sweetening properties to 6 g of sucrose (approximately 1 teaspoon of sucrose), applies”.
Comment 3.13:
- What do you mean exactly with ‘’Food quality’’? indicate parameters
Answer 3.13:
Food quality is an abstract term, for which each consumer has their own interpretation based on the different perceptions of product attributes (Ophuis, P.A.O. and Van Trijp, H.C., 1995. Perceived quality: A market driven and consumer oriented approach. Food quality and Preference, 6(3), pp.177-183.). We used „food quality” in this manner, as a general phrase.
Comment 3.14:
- Ref 46: check if it relevant to the statement in the text, if not necessary and representing national data, replace with article in English; check all other references
Answer 3.14:
The mentioned reference is not necessary, therefore we removed it according to your recommendation. Thank you for noticing this.
Round 2
Reviewer 3 Report
Comments and Suggestions for Authors
Thank you altering the manuscript according to suggestions.